# Proteomic Profiling of Thigh Meat at Different Ages of Chicken for Meat Quality and Development

**DOI:** 10.3390/foods12152901

**Published:** 2023-07-30

**Authors:** Jian Zhang, Xia Chen, Jing Cao, Cheng Chang, Ailian Geng, Haihong Wang, Qin Chu, Zhixun Yan, Yao Zhang, Huagui Liu

**Affiliations:** Institute of Animal Husbandry and Veterinary Medicine, Beijing Academy of Agriculture and Forestry Sciences, Beijing 100097, China; zjcau@126.com (J.Z.); chenxia_91@163.com (X.C.); caojing2046555@163.com (J.C.); changeng02@163.com (C.C.); ailiangengcau@126.com (A.G.); haioulantian@126.com (H.W.); chuqinsd@163.com (Q.C.); yanzhixun2008@sina.com (Z.Y.); duguyimeng1@126.com (Y.Z.)

**Keywords:** Beijing-You chicken, thigh meat, protein profile, free amino acid, intramuscular fat

## Abstract

Chicken age contributes to the meat characteristics; however, knowledge regarding the pathways and proteins associated with meat quality and muscle development are still scarce, especially in chicken thigh meat. Hence, the objective of this study was to elucidate the intricate relationship between these traits by liquid chromatography mass spectrometry at three different ages. A total of 341 differential expressed proteins (DEPs) were screened out (fold change ≥ 1.50 or ≤0.67 and *p* < 0.05) among 45 thigh meat samples (15 samples per age) of Beijing-You chicken (BYC), collected at the age of 150, 300, or 450 days (D150, D300, and D450), respectively. Subsequently, based on the protein interaction network and Markov cluster algorithm (MCL) analyses, 91 DEPs were divided into 26 MCL clusters, which were associated with pathways of lipid transporter activity, nutrient reservoir activity, signaling pathways of PPAR and MAPK, focal adhesion, ECM-receptor interaction, the cell cycle, oocyte meiosis, ribosomes, taurine and hypotaurine metabolism, glutathione metabolism, muscle contraction, calcium signaling, nucleic acid binding, and spliceosomes. Overall, our data suggest that the thigh meat of BYC at D450 presents the most desirable nutritional value in the term of free amino acids (FAAs) and intramuscular fat (IMF), and a series of proteins and pathways associated with meat quality and development were identified. These findings also provide comprehensive insight regarding these traits across a wide age spectrum.

## 1. Introduction

The age of chicken is one of the leading important factors associated with chicken growth performance, meat organoleptic attributes, and nutrition characters [1,2,3]. With the improvement of living conditions and health consciousness, the preference of consumers is shifting to older chicken because they believe that the older the bird, the better the flavor and nutrition [4]. As a consequence, the age of chicken has attracted great attention in recent years. In our previous study, we found that the age of the bird can affect both fatty acid composition and related metabolism in breast meat [5]. Moreover, for different types of chickens (commercial or indigenous), due to the different individual growth rates, the various marketable age has been suggested to be a significant factor influencing meat quality via studies integrating GC-MS and LC-MS technologies [6]. The results suggest that the Cobb broiler is characterized by a high expression of amino acids and derivatives. In contrast, Beijing-You chicken (BYC) features abundant expression of certain fatty acids and taurine.

Recently, intramuscular fat (IMF), free amino acids (FAAs), and peptides have been found to be associated with meat flavor, color, juiciness, and tenderness [7,8]. Farmer [9] reported that amino acids make an important contribution to meat smell. Zotte et al. [10] showed that Polverara, an indigenous purebred chicken, possesses a better nutritional meat quality than a commercial hybrid due to its better amino acid profile. Furthermore, published studies, mostly focused on the breast meat of birds of less than 150 days of age, have shown that the contents of IMF and FAAs in poultry meat are influenced by bird age [3,11,12]. For instance, in our previous study [12], a proteomic strategy was applied in chicken breast muscle to evaluate the protein expression profiles at three physiological stages from the age of 90 days to 150 days; the results demonstrated that the developmental stages have a great effect on the deposition of IMF through certain important pathways, including the ECM receptor and the focal adhesion pathway. In addition, Ge et al. [3] performed an HPLC-QTRAP-MS-based metabolomics approach to identify the metabolites at different developmental stages in birds of 56–120 days of age in chicken breast muscle and demonstrated that the age of the bird could affect the metabolites of amino acids. Nevertheless, until now, knowledge surrounding the mechanisms responsible for dynamic changes in meat quality and muscle development, especially for FAAs and IMF in thigh meat at the protein level across a wide laying-age spectrum, have remained scarce.

Accordingly, Beijing-You chicken (BYC), protected by geographical indications of agricultural products in China since 2020, was used to explore the dynamic changes in protein profiles for meat quality and muscle development with age increasing in this study. Furthermore, the differential expressed proteins (DEPs) and pathways related to FAAs and IMF were identified across a wide laying-age spectrum. Overall, we identified the protein expression profiles in thigh meat of BYC at three different post-laying ages (150, 300, and 450 days) through a TMT-LC-MS/MS method, representing the first-laying age (sexual maturation age when the first egg was seen, D150), medium-laying age (D300), and later-laying age (culling age, D450), respectively. In order to eliminate feeding interference, all hens were fed with the same type of diet and under the same conditions. Furthermore, the results of TMT-LC-MS/MS were further confirmed by the parallel reaction monitoring (PRM) method with eight selected candidate proteins. Taken together, the information derived from this study can help in better understanding the mechanisms underlying meat quality and muscle development.

## 2. Materials and Methods

### 2.1. Ethics Statement

The animal study was reviewed and approved by the Science Research Department of the Institute of Animal Husbandry and Veterinary Medicine, Beijing Academy of Agriculture and Forestry Sciences (Beijing, China) with the following reference number: BAAFS-IAHVM20190116.

### 2.2. Animals and Samples Preparation

A total of ninety BYC hens were obtained from the Institute of Animal Husbandry and Veterinary Medicine, Beijing Academy of Agriculture and Forestry Sciences. The birds were raised in an environmentally controlled room with three-floor pens and a three-phase feeding system (0–49, 49–120, and 120–450 d), divided into three groups (30 chickens per group) in this study. The crude protein and metabolizable energy for these three-phase diets were 19.00% and 11.91 MJ/kg, 15.07% and 11.20 MJ/kg, and 15.51% and 11.08 MJ/kg, respectively. Diets and water were available ad libitum throughout the whole rearing period. At each post-laying age (D150, D300, and D450), 15 birds were randomly selected, after a 12 h overnight fast. Then, the birds were electrically stunned and killed via exsanguination. The left thigh samples (biceps femoris) were collected in liquid nitrogen and stored at −80 °C until TMT-LC-MS/MS was applied and the right thigh samples were stored at 4 °C before determining the characteristics of meat quality.

### 2.3. Meat Quality Characteristics

The dry matter content of the thigh meat was obtained based on the percentage of dried meat weight to fresh meat weight through a freeze-dryer (Millrock Technology, Kingston, NY, USA). IMF was extracted using petroleum ether in a Soxhlet apparatus according to the method of Zerehdaran et al. [13]. The composition of free amino acids and certain peptides in thigh meat was measured as described by Chen et al. [6] using an amino acid analyzer (L-8900, Hitachi Ltd., Tokyo, Japan).

### 2.4. Protein Extraction, Digestion and TMT Labeling

The meat samples were prepared following our previous published protocols [5]. In brief, 1.5 mL of protein lysis buffer (8 M urea, 1% SDS) was used to suspend the powder of meat samples (100 mg). Then, after being incubated on ice for 30 min and sonicated for 2 min, the supernatant was collected and quantified through a BCA Protein Assay Kit (Thermo Scientific). Lastly, 9 pools (5 samples per pool and 3 pools per age stage) with the same concentration were obtained according to protein content. After being digested at 37 °C overnight with trypsin (the mass ratio of trypsin-to-protein at 1:50), the peptides were labeled with 10-plex TMT reagents (Thermo Fisher Scientific) and multiplex labeled samples were obtained.

### 2.5. LC-MS/MS Analysis and Protein Identification

The protein identification and quantification data were further analyzed, as described previously [5]. To increase the proteomic depth, Vanquish Flex binary UHPLC chromatography (Thermo, Waltham, MA, USA) with the ACQUITY UPLC BEH C18 Column (1.7 μm, 2.1 mm × 150 mm, Waters, Milford, MA, USA) was applied to fractionate the resuspension of TMT-labeled peptides. At a flow rate of 200 μL/min, peptides were first separated with a gradient of buffer B (5 mM ammonium hydroxide solution containing 80% acetonitrile, pH 10), and then 15 fractions pooled were obtained from 30 fractions per sample. A quadrupole orbitrap mass spectrometer (Q Exactive HF-X) coupled to an Easy nLC 1200 (Thermo Fisher Scientific) were applied in this study. The data-dependent acquisition mode (DDA) was selected to automatically switch between full-scan MS and MS/MS acquisition. The resolution of the MS spectra (350–1500 *m*/*z*) was 60,000 at *m*/*z* 100, and the MS/MS resolution was set at 45,000. The AGC, maximum injection time, and dynamic exclusion were set at 2.0 × 105, 96 ms, and 30 s, respectively.

The raw files of LC-MS/MS were processed using the ProteomeDiscoverer software (Thermo Scientific, version 2.4) to identify the protein against the database UniProt-gallus gallus-34930-20200807.fasta (released in August 2020 and including 34,930 sequences). Through the iProX partner repository, the mass spectrometry proteomics data have been deposited into the ProteomeXchange Consortium (PXD031857), as described previously [14].

### 2.6. LC-PRM/MS Analysis

In order to validate the TMT proteomics data, further liquid chromatography–parallel reaction monitoring MS (LC-PRM/MS) analysis was applied as described by Zhang et al. [5]. In brief, for each target protein, 1–3 unique peptides with high intensity and confidence were applied to optimize the methods of PRM analysis using a Q Exactive HF-X mass spectrometer (Thermo Fisher Scientific). The signal intensities of individual peptide sequences were obtained by Skyline 4.1 (MacCoss Lab, University of Washington).

### 2.7. Statistical Analysis and Bioinformation Analysis

Meat quality characteristics were analyzed by the general linear model procedure of SAS (version 9.2, SAS Institute Inc., Cary, NC, USA). The main effect was set as the three different post-laying ages (D150, D300, and D450), and significant differences between LSmeans were identified through Tukey’s method (*p* < 0.05). DEPs were screened out based on the thresholds of fold change (FC) ≥ 1.50 or ≤0.67 and *p*-values < 0.05. GO (BLAST2GO, version 2.5.0) and KEGG (version 2018) were used to annotate all identified proteins. The enrichment analyses of GO and KEGG were carried out with Fisher’s exact test. In addition, FDR correction for multiple testing was also performed. The enrichment analyses of GO and KEGG pathways were conducted at a significance level of *p* < 0.05 and protein–protein interaction (PPI) network analysis were performed through STRING (version 11.5).

## 3. Results

### 3.1. Characteristics of Thigh Meat Quality during Growth

The quality characteristics (live weight, crude protein, IMF, FAAs, and peptides) of thigh meat samples at D150, D300, and D450 are presented in Table 1. In general, bird live weight, weight and percentage of thigh meat, dry matter, and IMF exhibited the greatest values at D450, followed by D300, and then D150 (*p* ≤ 0.01). However, crude protein presented the opposite trend. In addition, there were many compounds (FAAs or peptides) which were detected differently at various post-laying ages. Among these free amino acids, taurine showed the highest value, accounting for more than 57% of the total amino acids, and presented higher values at D450 than that at D150 and D300, which did not differ. In addition, aspartic acid and total amino acids at D450 were also higher than that at D150 and D300, while carnosine, glycine, valine, leucine, phenylalanine, alanine, and threonine at D300 and D450, which were not different from each other statistically (*p* > 0.05), were higher than at D150 (*p* < 0.05). These data indicated that complicated changes in thigh meat quality were constantly occurring with the increase in the bird post-laying age.

### 3.2. Proteomic Expression Profiling of Thigh Muscle

At a false discovery rate (FDR) of ≤0.01, a total of 73,624 spectra, 35,421 peptides, and 4465 proteins were identified, respectively (Appendix A). The detected proteins were extremely distinguishable between the different post-laying age samples by analyses of correlation and principal components (Appendix A). We applied GO term analysis to elucidate the function of these identified proteins based on biological process (BP), cellular component (CC), and molecular function (MF), respectively (Appendix A). Furthermore, the results of KEGG functional annotation illustrated that these proteins were primarily related to the pathways of transport and catabolism, translation, signal transduction, carbohydrate metabolism, and the endocrine system (Appendix A).

### 3.3. Functional Enrichment Analysis of DEPs

A total of 341 DEPs were identified, and the distributions of these DEPs between different post-laying age groups were visualized by volcano plots (Appendix A). A comparative analysis of protein expression showed that 145, 10, and 181 proteins were significant upregulated and 36, 56, and 90 proteins were downregulated between the different comparative groups of D150 vs. D300, D300 vs. D450, and D150 vs. D450, respectively (Appendix A). In addition, based on the *p* values, the top 10 DEPs of each comparison group are listed in Appendix A, which might contribute to chicken thigh meat development and meat quality changes during growth.

### 3.4. Subcluster Analysis of DEPs

To visualize the expression patterns, a hierarchical cluster analysis was applied based on the protein abundance data of 341 DEPs. In general, the expression patterns were more similar between D300 and D450 (Appendix A). Additionally, these 341 DEPs could be further subdivided into 5 distinct subclusters during growth according to their expression pattern (Figure 1 and Appendix A). Subcluster 1 (68 proteins) showed no obvious differences in protein abundance between D150 and D300; however, these were upregulated significantly from D300 to D450 (Figure 1A). The proteins, enriched in Subcluster 1, were mostly related to the cell cycle (CDC27, CDKN2C, and TGFB1), fucosyltransferase (FUT6), and ECM-receptor interaction (LAMB4 and COL4A1). In addition, the largest subcluster (Subcluster 2, including 208 proteins) illustrated a sharp downregulation from D150 to D300, while no obvious difference in protein abundance was detected between D300 and D450 (Figure 1B). Thereinto, proteins associated with the spliceosome (hnRNPA1, hnRNPU, SRSF7, SRSF3, SNRPD2, TRA2B, and RBMXL1), taurine and hypotaurine and glutathione metabolism (GGT5, GADL1, and GSTT1), the cell cycle and oocyte meiosis (YWHAZ, YWHAQ and YWHAH), and the PPAR signaling pathway (APOC3 and APOA1) were identified. In contrast, Subcluster 3 (50 proteins) exhibited a protein abundance which was upregulated steadily from D150 to D300, while no obvious difference in protein abundance was detected between D300 and D450 (Figure 1C). In Subcluster 3, the proteins were related mostly to the metabolism of arginine and proline, vitamin B6, signaling pathway of calcium and MAPK were identified (AGMAT, CKMT2, PNPO, STIM1, TNNC1, and HSPB1). Moreover, it is worth noting that Subcluster 4 (8 proteins) and Subcluster 5 (7 proteins) were characterized by a great difference in protein abundance at D300 compared to those at D150 and D450 (Figure 1D,E). Certain representative proteins (HSP90AB1, COL1A2, COL3A1, and BRT-2) with functions of progesterone-mediated oocyte maturation, adrenergic signaling in cardiomyocytes, AGE-RAGE signaling pathway in diabetic complications, and ECM-receptor interaction were detected in Subcluster 4 and Subcluster 5.

### 3.5. Function Analysis of DEPs Based on Age Stages

According to the results of the above five distinct protein expression patterns during growth, 258 proteins, coming from Subcluster 2 (208 proteins) and Subcluster 3 (50 proteins), were regarded as DEPs of D150 discriminated from the other two stages (D300 and D450). Similarly, proteins of Subcluster 4 (8 proteins) and Subcluster 5 (7 proteins) comprised the 15 special DEPs for D300 compared with D150 and D450. In contrast, proteins of Subcluster 1 (68 proteins) were considered as the noted DEPs for D450, different from D150 and D300. Functional enrichment analyses of GO and KEGG pathway were carried out to further obtain the functional dynamic change in the DEPs due to age stage. A total of 247, 156, and 178 GO terms were enriched (*p* < 0.05) for D150, D300, and D450, respectively (Appendix A), and the top 10 categories in GO terms are shown in Figure 2. The results of KEGG pathway enrichment analysis illustrated that 2, 2, and 5 KEGG pathways were enriched (*p* < 0.05) for D150, D300, and D450, respectively (Appendix A). The top 12 KEGG pathways of D150, D300, and D450 are illustrated in Figure 3.

### 3.6. Interaction Networks of DEPs

To further explore the PPI networks altered in thigh meat with increasing age, the web-tool STRING 11.5 (http://string-db.org) (accessed on 23 February 2023).was applied to construct these DEPs’ interaction networks. Following this, 224 of 341 DEPs were detected in STRING, and 91 proteins were eventually screened out with high confidence interactions (CI > 0.70) and exclusion of the nodes disconnected in the network. In addition, the above network was further clustered based on Markov cluster algorithm (MCL) clustering. Finally, a total of 26 MCL clusters were obtained (Figure 4) when the inflation parameter was set at 2.0. The largest cluster, including eight proteins (APOA4, C8A, C8B, FETUB, HABP2, ORM1, SERPINA3, and SERPINA4), was associated with extracellular space and region. We also found interaction networks for glutathione metabolic processes (GSTT1, GSTA3, GPX1, and GLRX), muscle contraction (TNNT3, TNNC1, MYH1F, TPM2, and MYBPC1), cardiac muscle contraction (ATP1B1 and ATP1B4), the cell cycle and oocyte meiosis (YWHAQ, YWHAZ, and YWHAH), the MAPK signaling pathway (CASP3, HSPB1, ANXA5, ANXA13, NMT1, PAK2, and VIM), the activity of the nutrient reservoir and lipid transporter (APOA1, APOV1, VTG2, and VTG3), annexin repeat (ANXA8L1, S100A10, and FHL2), spliceosomes (hnRNPA0, hnRNPA1, hnRNPA3, hnRNPU, RBMXL1, SRSF3, and TRA2B), nucleic acid binding (HIST1H46, HISTH1, H3-I, H2AFX, SSRP1, ENSGALP00000027025, and DDX21), oxidoreductase activity (COX6A1, NDUFA13, NDUFB3, UQCR10, and IDH3B), and ribosomes (RPL29 and ENSGALP00000041962), etc.

### 3.7. PRM Validation of TMT-Based Results

In the present study, eight target proteins were selected to validate the results of TMT based on LC-PRM/MS quantitative analysis, including two proteins (TNNC1 and CKMT2) related to muscle contraction and energy transduction, two proteins (APOA4 and APOA1) associated with lipid transport and storage, and four proteins (COL1A1, COL1A2, LUM, and VIM) contributing to the structural integrity of the extracellular matrix and intermediate filament organization, respectively. In general, PRM results support the plausibility and reliability of the TMT data, exhibiting a good correlation with the proteomics data (Appendix A).

## 4. Discussion

### 4.1. The Proteins and Pathways Related to FAAs and Peptides

Free amino acids (FAAs), acting as one of the meat quality traits, has been demonstrated to make an important contribution to meat flavor [15]. In this study, a total of 15 FAAs or peptides were identified and expressed differently at various post-laying ages in chicken thigh meat. In general, a significant rising trend in these compounds was found in thigh meat with increasing age. The most detected FAAs or peptides of D450 were higher than that of D150 (*p* < 0.05), except for a few compounds, such as anserine, serine, and arginine. In particular, aspartic acid, carnosine, taurine, and total FAAs reached the maximum at D450. These results might partly explain the reason why the consumers most prefer to chicken with an older age, especially in Asian countries [4].

Meister [16,17] has proved that γ-glutamyltransferase (GGT) is responsible for the transport of amino acids, glutamine formation, and glutathione metabolism through the γ-glutamyl cycle. In the present study, we found gamma-glutamyltransferase 5 (GGT5), glutathione S-transferase theta-1 (GSTT1), glutathione S-transferase (GSTA3), and glutamate decarboxylase-like 1 (GADL1), classified into Subcluster 2 (Figure 1B and Appendix A). Furthermore, certain proteins, including GGT5, GADL1, GSTT1, GSTA3, and GPX1, were found to be involved in taurine and hypotaurine metabolism (gga00430) and glutathione metabolism (gga00480) by pathway enrichment analysis. Additionally, in the present study, both the total amino acids and taurine presented the highest values at D450. Hence, we speculated that the expression level of the abovementioned proteins might have a negative effect on taurine and glutathione metabolism. The KEGG enrichment analysis result, based on the 258 DEPs specific to D150 (Figure 3A), and the protein interaction network (Figure 4) further confirmed that GGT5, GSTT1, GPX1, and GSTA3 play a critical role on taurine and glutathione metabolism. On the other hand, AGMAT and CKMT2 were found to be enriched in arginine and proline metabolism (gga00330), classified into Subcluster 3 (Figure 1C and Appendix A). Taken together, with increasing age, the concentrations of FAAs showed a significantly rising trend generally, especially for the first 150 days since laying started. Thus, in terms of FAAs and peptide concentrations, the thigh meat of BYC at D450 presented a more desirable nutritional value.

### 4.2. The Proteins and Pathways Related to IMF

Previous studies have indicated that, different from mammals, chickens’ liver is the main organ for lipid biosynthesis [18], which is further transported to target tissues through the peripheral vascular system, in the form of triglyceride-rich very low density lipoprotein [19]. Apolipoproteins, such as APOA1, APOA-Ⅳ, and APOC3, are responsible for lipid transport [20]. Vitellogenins (VTGs) are characterized by multidomain proteins, belonging to a large family of lipid transfer genes, and account for the transportation of lipids to the ovary [21], which can be explained by the attributes of lipid binding and transport [22]. In this study, the proteins APOC3, APOA1, STMN1, HSPB1, PAK2, TGFB1, and CASP3 were significantly enriched in the signaling pathways of PPAR (gga03320) and MAPK (gga04010), respectively. Additionally, COL4A3BP, ATP8B3, APOA1, APOA-IV, VTG1, VTG2, and VTG3 were predominantly enriched in lipid transporter activity (GO:0005319). Further GO enrichment analysis revealed that three common proteins, VTG1, VTG2, and VTG3, were shared by the terms of lipid transporter activity (GO:0005319) and nutrient reservoir activity (GO:0045735), suggesting that these two GO terms were interacting with each other. These results agreed with the finding of previous studies [5,21], suggesting that the VTG family of proteins work as the nutrition reserve pool and are responsible for the regulation of lipid transportation. Furthermore, we found that the expression of VTGs and APOV1 were lowest at D150 and belonged to Subcluster 3 (Figure 1C and Appendix A); the protein interaction network is shown in Figure 4. Liu et al. [11] systematically investigated IMF accumulation in chicken breast meat at post-hatching ages of 1, 56, 98, and 140 days, reporting that IMF decreased dramatically from Days 1 to 56, then stepped increased from Days 56 to 140. In our study, IMF increased with increasing post-laying age in chicken thigh meat. Additionally, the content of IMF dramatically increased by more than 88% from D150 to D300, whereas it only increased by approximately 40% from D300 to D450, respectively. This result suggested that the relative speed of IMF deposition from D150 to D300 was two times larger than that from D300 to D450 (Table 1). Taken together, a significant rising trend with age was identified in the content of IMF, especially in the first 150 days since laying started, which was supported by the results of GO enrichment analysis which suggested that nutrient reservoir activity was one of the significant GO terms enriched for D150 (Figure 2A). Furthermore, the above results agreed with the reports of previous studies [11,13] suggesting that fat deposition increases in chicken meat with increases in the rearing period and the proteins of APOV1, VTG2, and VTG3 play a critical role in lipid transport.

On the other hand, GO enrichment analysis revealed that COL1A1, COL1A2, COL3A1, COL4A1, COL28A1, LAMB4, APOA4, HABP2, ORM1, C8A, FETUB, SERPINA4, SERPINA3, and C8B were involved in the categories of extracellular matrix structural constituent (GO:0005201) and extracellular region (GO:0005576), respectively. Additionally, pathway enrichment analysis showed that four common proteins, COL1A2, COL4A1, LAMB4, and THBS1, were shared by the ECM-receptor interaction (gga04512) and focal adhesion (gga04510) terms, suggesting that these two pathways were interacting with each other. Furthermore, we found that COL1A1, COL1A2, and COL3A1, classified into Subcluster 5, presented the highest abundance values at D300 (Figure 1E and Appendix A). However, COL4A1 and LAMB4 showed the highest abundance values at D450, classified into Subcluster 1 (Figure 1A and Appendix A). These results are supported by the fat deposition continuously increasing with increasing age (Table 1) and the results of GO enrichment analysis, which proposed that the extracellular structure organization, extracellular matrix organization, and collagen trimer were the significant GO terms enriched for the post-laying age of D300 (Figure 2B). The above results suggest that the dominant proteins associated with the extracellular matrix structural constituent varied with the rearing period, accompanied by IMF deposition.

Taken together, it was speculated that the IMF transport and deposition process is accompanied with the processes of focal adhesion and ECM-receptor interaction. These speculations are supported by the data published in previous studies [7,23], which reported that cell junctions of focal adhesion and ECM-receptor interaction play critical roles in the IMF accumulation which accompanies the function of the PPAR signaling pathway. In addition, the pathways of ECM-receptor interaction and focal adhesion were also proven to be associated with the deposition of IMF in chicken breast meat at different developmental stages at 90–150 days of age in our previous study [12]. Hence, further study is needed to explore the intricate relationship between the cell junctions and IMF accumulation, which might help to provide insights into extracellular matrix structural functions, transport and accumulation of IMF, and underlying mechanisms.

### 4.3. The Proteins and Pathways Related to Muscle Development and Contraction

As both the thigh muscle absolute weight and the weight percentage relative to the bird live weight significantly increased with increasing age (Table 1), it is not surprising that several critical proteins involved in muscle development were identified in the present study. For example, RPL29, RPL34, ATP1B1, ATP1B4, ATP1A1, BRT-2, TNNC1, COX6A1, MYL12A, and ARPC3 were found to be enriched in the ribosome (gga03010), cardiac muscle contraction (gga04260), and regulation of the actin cytoskeleton (gga04810). These results agreed with the finding that functional proteins related to the ribosome and muscle contraction were expressed differently between various ages [11,24]. In addition, the annexin (ANX) family is a group of multifunctional calcium-dependent membrane phospholipid binding proteins, proposed to have central roles in many biological processes including calcium signaling, membrane organization, and membrane traffic [25]. In addition, Liu et al. [26] suggested that S100A10 is a functional gene involved in the development and lipid transport of adipocytes. In the present study, several proteins of the ANX family, such as ANXA5, ANXA13, and ANXA8L1, were identified, as well as their binding partner S100A10. Furthermore, we found that ANXA5, ANXA13, S100A10, RPL29, RPL34, ATP1B1, and ATP1B4, classified into Subcluster 2, presented the highest abundance values at D150 (Figure 1B and Appendix A). However, TNNC1 and ANXA8L1 showed the lowest abundance values at D150 and were classified into Subcluster 3 (Figure 1C and Appendix A), which agreed with the highest muscle growth rate at D150 in the present study (Table 1). Furthermore, we found the interaction network for ribosomal proteins (RPL29 and RPL34), sodium/potassium-transporting ATPase subunit proteins (ATP1B1 and ATP1B4), muscle contraction (TNNC1, TNNT3, etc.), and annexin proteins (ANXA8L1, ANXA13, and S100A10), as presented in Figure 4.

In addition, accumulated works of evidence have demonstrated that histones have a great effect on a wide range of functions, such as chromatin structure modulation and gene regulation, through their post-translational modifications [27,28]. Myogenesis has been reported to be controlled by sequential chromatin regulation based on the selection and modification of appropriate histone [29]. Histone family proteins have also been found to express differently in breast meat between the ages of hatching and post-hatching [11]. In our study, several proteins (HIST1H46, HISTH1, H2A, H2AFX, H3-I, SSRP1, and DDX21) involved in nucleic acid binding (GO:0003676) and protein heterodimerization activity (GO:0046982) were identified by GO enrichment analysis. A total of four shared proteins (HIST1H46, H2AFX, H3-I, and H2A) were found to be enriched in the above two GO terms, which was supported by the protein interaction network (Figure 4). Additionally, we found that HIST1H46, HISTH1, H2AFX, H3-I, and H2A were classified into Subcluster 2 (Figure 1B and Appendix A), which agreed with the highest muscle growth rate presented at D150 as well (Table 1). Based on the above results, we surmised that RPL29, RPL34, ATP1B1, ATP1B4, TNNC1, ANXA5, ANXA13, ANXA8L1, S100A10, and the histone-related proteins were responsible for the muscle development, contraction, and meat quality, which is consistent with previous research [30].

### 4.4. The Proteins and Pathways Related to Follicle Development

Growing evidence shows that the 14-3-3 family of proteins (YWHA) are related to cell growth and differentiation, negative regulation of apoptosis, and cell cycle regulation [31,32]. In avian and mouse reproduction studies, the 14-3-3 family of proteins have been confirmed to be associated with follicle development and oocyte maturation [33]. It is well known that reproduction from follicle maturation to ovulation is continuously taking place during the laying stage in the avian. Therefore it is not surprising that several proteins including YWHAZ, YWHAQ, YWHAH, CDC27, CDKN2C, and TGFB1 were found to be enriched in the signaling pathways of the cell cycle (gga04110) and oocyte meiosis (gga04114), respectively. Furthermore, three shared proteins (YWHAZ, YWHAQ, and YWHAH) were enriched in the above two pathways and were clustered in a protein interaction network (Figure 4). Additionally, we found that these three 14-3-3 proteins, classified into Subcluster 2, presented the highest abundance values at D150 (Figure 1B and Appendix A). The high expression levels of 14-3-3 proteins have been reported to repress the transition from G2 to M phase [34], and lower concentrations of 14-3-3 proteins appeared in mature oocytes than in immature oocytes [35]. Furthermore, Eisa et al. [36] demonstrated that 14-3-3 proteins have a great effect on sustaining meiotic arrest and on the regulation of the maturation of oogenesis and the oocyte. All these pieces of evidence suggest that the status of oocyte maturation at D150 was less than that at D300 and D450, which could explain why the highest abundance values of 14-3-3 proteins were detected at D150 (Figure 1B) and why the cell cycle pathway was significantly enriched based on the 68 specific DEPs of D450 (Figure 3C). Thus, we speculated that 14-3-3 proteins play a pivotal role in follicle development and could work as biomarkers to evaluate the potential bird laying ability, which is supported by the study of Shen et al. [33], who reported that follicle development during different reproductive stages in chickens may be affected by 14-3-3 proteins in serum.

### 4.5. The Proteins and Pathways Related to the Spliceosome

It is widely believed that alternative splicing (AS) is related to gene spatial–temporal expression and exerts important physiological functions, such as adaptive immunity, muscular function, neurogenesis and neuron maturation, gametogenesis, etc. [37]. Heterogeneous nuclear ribonucleoprotein complexes (hnRNPs), acting as the core components of the spliceosome, are involved in numerous RNA-related activities [38]. Furthermore, Gallego-Paez et al. [37] indicated that the multifunctional RNA-binding protein CUGBP Elav-like family member 2 (CELF2), regulates several stages of RNA processing and is expressed broadly in developing stages. In this study, we found that hnRNPU, hnRNPA1, CELF2, TRA2B, SRSF3, SRSF7, RBMXL1, and SNRPD2 were significantly enriched in the spliceosome pathways (gga03040), regulation of alternative mRNA splicing (GO:0000381), and regulation of mRNA splicing (GO:0048024), respectively. Furthermore, the proteins mentioned above derived from Subcluster 2 (Figure 1B and Appendix A), which might contribute significantly to much stronger physiological changes at D150, representing the sexual maturation age when the first egg is seen. These results were further confirmed by the protein interaction network (Figure 4), as well as the results of enrichment analysis suggesting that the negative regulation of lymphocyte differentiation, negative regulation of leukocyte differentiation, and the spliceosome were significantly enriched based on the 258 specific DEPs of D150 (Figure 3A). The splicing regulators of hnRNP LL and CELF2 have been shown to be upregulated to activate T-cells [39,40], which is in agreement with our study. Therefore, we suggest that hnRNPU, hnRNPA1, CELF2, and TRA2B play critical roles in RNA-related activities during the post-laying period.

## 5. Conclusions

In this study, the protein expression profiles of chicken thigh meat across a wide post-laying age spectrum were characterized by TMT-LC-MS/MS-based quantitative proteomic analysis. A significant rising trend was detected in the content of most detected FAAs and IMF, and the protein expression profiles of D150 were vastly different from D300 and D450. As a result, a better desirable nutritional value, in the term of the FAAs and IMF, was shown at D450. Certain important proteins which contributed significantly to the meat quality and muscle development were identified, which were associated with ribosome, muscle contraction, calcium signaling, nucleic acid binding, taurine and hypotaurine metabolism, glutathione metabolism, lipid transporter activity, nutrient reservoir activity, the signaling pathways of PPAR and MAPK, and focal adhesion, as well as ECM-receptor binding. In addition, YWHA, hnRNPs, CELF2, and TRA2B had a strong effect on RNA-related activities and regulating the maturation of oogenesis and oocytes during the post-laying period. All in all, our results disclose that the chicken age exerted great effects on meat quality through certain important proteins and pathways, which could help in better understanding the biological process involved in chicken thigh meat quality formation and muscle development during the post-laying period.

## Figures and Tables

**Figure 1 foods-12-02901-f001:**
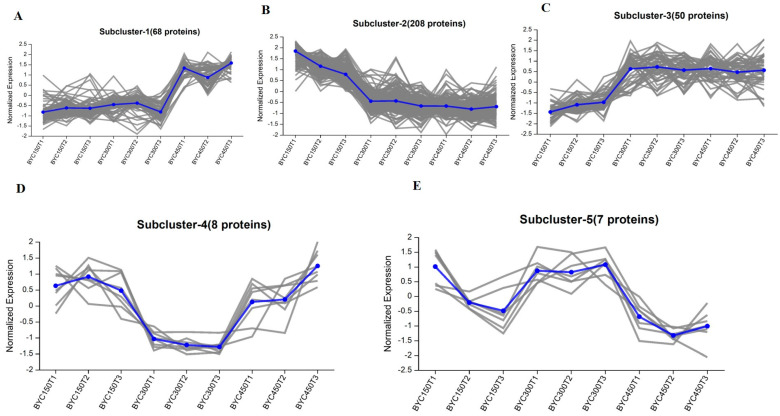
Subcluster analysis of protein abundance trends in the 341 DEPs during growth (D150, D300, and D450) in thigh meat. (**A**) Subcluster 1 (68 proteins) protein abundance exhibited no significant difference from D150 to D300, then were upregulated sharply from D300 to D450. (**B**) Subcluster 2 (208 proteins) protein abundance at D150 was higher than that at D300 and D450. (**C**) Subcluster 3 (50 proteins) protein abundance at D150 was lower than that at D300 and D450. (**D**) Subcluster 4 (8 proteins) protein abundance at D300 was lower than that at D150 and D450. (**E**) Subcluster 5 (7 proteins) protein abundance at D300 was higher than that at D150 and D450. Each gray line in the graph represents a protein, and the blue line represents the average expression level of all proteins in the individual Subcluster.

**Figure 2 foods-12-02901-f002:**
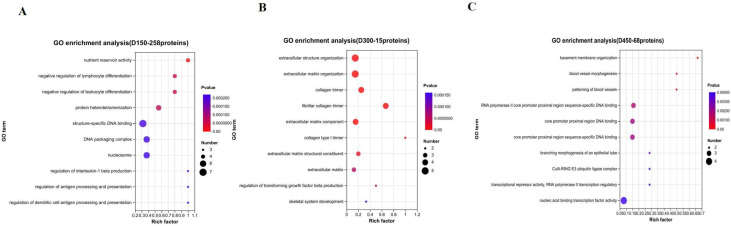
The top 10 categories in GO functional enrichment analysis during growth. (**A**) D150; (**B**) D300; (**C**) D450.

**Figure 3 foods-12-02901-f003:**
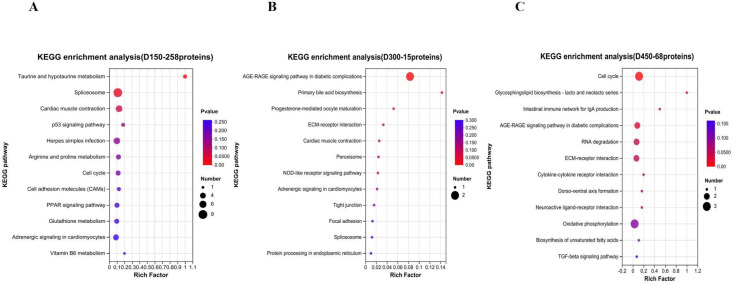
The top 12 categories in KEGG pathway enrichment analysis during growth. (**A**) D150; (**B**) D300; (**C**) D450.

**Figure 4 foods-12-02901-f004:**
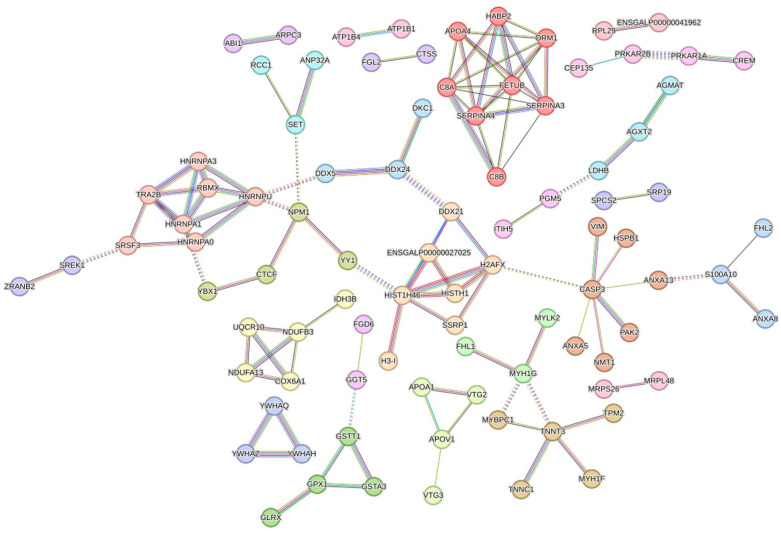
The protein–protein interaction network of 341 differentially expressed proteins during the post-laying age. In this network, nodes represent proteins, and different colors represent the predicted different clusters.

**Table 1 foods-12-02901-t001:** Characteristics of chicken thigh meat quality at three different post-laying ages (mean ± SD, *n* = 15) ^1^.

Traits ^2^	D150	D300	D450
Live weight (g)	1344 ± 63 ^c^	1951 ± 266 ^b^	2228 ± 294 ^a^
Thigh muscle weight (g)	55.7 ± 10.2 ^c^	99.5 ± 14.6 ^b^	134.2 ± 21.3 ^a^
Percentage of thigh weight (%)	8.3 ± 1.5 ^c^	10.3 ± 1.4 ^b^	12.0 ± 0.9 ^a^
IMF (%)	4.4 ± 0.8 ^c^	8.3 ± 1.2 ^b^	11.7 ± 2.3 ^a^
Crude protein (%)	79.18 ± 3.00 ^a^	72.28 ± 3.06 ^b^	61.83 ± 4.78 ^c^
Dry matter (%)	25.17 ± 1.15 ^c^	30.17 ± 1.28 ^b^	32.53 ± 1.80 ^a^
Ans (mg/g)	3.95 ± 0.43 ^a^	2.61 ± 0.32 ^b^	2.58 ± 0.36 ^b^
Car (mg/g)	0.39 ± 0.20 ^b^	1.95 ± 0.72 ^a^	2.17 ± 0.83 ^a^
Ser (mg/g)	0.09 ± 0.02 ^a^	0.05 ± 0.01 ^c^	0.07 ± 0.02 ^b^
Tau (mg/g)	2.17 ± 0.31 ^b^	2.18 ± 0.42 ^b^	2.55 ± 0.47 ^a^
Asp (mg/g)	0.07 ± 0.03 ^b^	0.08 ± 0.04 ^b^	0.11 ± 0.03 ^a^
Thr (mg/g)	0.07 ± 0.03 ^b^	0.11 ± 0.07 ^a^	0.10 ± 0.04 ^a^
Ser (mg/g)	0.27 ± 0.10 ^a^	0.20 ± 0.06 ^b^	0.23 ± 0.05 ^ab^
Glu (mg/g)	0.25 ± 0.07 ^b^	0.32 ± 0.10 ^a^	0.28 ± 0.04 ^ab^
Gly (mg/g)	0.13 ± 0.05 ^b^	0.19 ± 0.06 ^a^	0.22 ± 0.03 ^a^
Val (mg/g)	0.03 ± 0.01 ^b^	0.04 ± 0.01 ^a^	0.04 ± 0.01 ^a^
Leu (mg/g)	0.03 ± 0.01 ^b^	0.04 ± 0.01 ^a^	0.04 ± 0.01 ^a^
Phe (mg/g)	0.019 ± 0.002 ^b^	0.022 ± 0.005 ^a^	0.023 ± 0.002 ^a^
Ala (mg/g)	0.04 ± 0.02 ^b^	0.07 ± 0.02 ^a^	0.06 ± 0.02 ^a^
His (mg/g)	0.02 ± 0.00 ^c^	0.03 ± 0.01 ^b^	0.04 ± 0.01 ^a^
Arg (mg/g)	0.12 ± 0.03 ^a^	0.06 ± 0.02 ^c^	0.08 ± 0.02 ^b^
Total amino acids (mg/g)	3.71 ± 0.36 ^b^	3.81 ± 0.52 ^b^	4.26 ± 0.50 ^a^

^1^ Values within a row followed by different superscript letters (a–c) differ significantly (*p* ≤ 0.05). ^2^ IMF: intramuscular fat; Ans: anserine; Car: carnosine; Ser: serine; Tau: taurine; Asp: aspartic acid; Thr: threonine; Ser: serine; Glu: glutamic acid; Gly: glycine; Val: valine; Leu: leucine; Phe: phenylalanine; Ala: alanine; His: histidine; Arg: arginine. D150: Day 150; D300: Day 300; D450: Day 450.

## Data Availability

The datasets presented in this study can be found in online repositories. The names of the repository/repositories and accession number(s) can be found below: ProteomeXchange Consortium via the iProX partner repository and PXD031857.

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
