# Peer review of "Proteomic Profiling of Thigh Meat at Different Ages of Chicken for Meat Quality and Development"

_foods, 2023, doi:10.3390/foods12152901_

Round 1

Reviewer 1 Report

Overall, the manuscript is well written. No major concerns found. My comments are given below.

·        It’s an interesting manuscript which provides the protein profile of chicken thigh meat. It seems parallel work to the previous paper reported by the same team. In this manuscript, authors used high-throughput MS and bioinformatics analyses to justify their findings. I appreciate the authors for their well written and well-presented manuscript.

·        Few grammatical errors must be rectified. Writing and English seems fine in general sense.

·        Around 4,000 proteins were identified from 35,000 peptides. Typically, it is a great number in MS. What fraction of theoretical proteins/peptides are missing in the MS analysis? Can we cover the entire theoretical proteins/proteins in the MS analysis?

·        I am curious to know the variability of protein profile in chickens from different regions/countries. Do you have any idea about that? Is there any comparative profile analysis? If so, cite the reference and discuss it in the manuscript.

·        Authors mentioned that “hnRNPU, hnRNPA1, 438 CELF2 and TRA2B play a critical role on the RNA-related activities during post-laying 439 period”. I would like to see some validation experiments (like western blot, or qPCR)

·        I found volcano plots and heatmap were in supplementary figures. It would be obvious that if you make them main figures.

Quality of English language looks fine

Reviewer 2 Report

The manuscript needs revision. Please refer to comments given in the text of reviewed attached file of the manuscript.

Reviewer 3 Report

The present form of the manuscript is well-written. I think that the topic concerning this submission should be interesting for readers.

Reviewer 4 Report

To Authors,

I congratulate you on the idea and the thoroughly conducted experiment, and the clear presentation of the results and conclusions. I believe that this is an original manuscript on the study of the protein profile for characteristics of chicken thigh meat at different ages, submitted for the special issue: Applications of Proteomics in Food Technology of the Foods and meets all the requirements for this type of submission. I only pay attention to minor spelling errors in English, such as the lack of double l signalling in a word. That's why I suggest a thorough spell check. In the Conclusions section, the final sentence, "Our discovery can help you better understand the biological process involved in chicken leg meat shaping the quality and development of muscles in the post-lay period." should be elaborated on how it can help with this better understanding?
    Many factors affect the characteristics of meat. However, the pathways and proteins associated with the effect of age on meat quality and muscle development have not been studied so far. The research results are, therefore, pioneering. The present study sought to elucidate the complicated relationship between these features using liquid chromatography-mass spectrometry. Proteomic studies included a total of 341 differentially expressed proteins (DEP) (fold change ≥ 1.50 or ≤ 0.67 and p < 0.05) from 45 thigh meat samples (15 pieces per age) of Beijing-You chicken (BYC) collected at the age of 150 years, respectively 300 or 450 days (D150, D300 and D450). The authors analyzed the protein interaction network and the Markov cluster (MCL) algorithm; The 91 DEPs were divided into 26 MCL clusters that were linked to lipid transporter activity pathways, nutrient reservoir activity, PPAR and MAPK signalling pathways, focal adhesion, ECM-receptor interaction, cell cycle, oocyte meiosis, ribosome, taurine and hypotaurine metabolism, glutathione metabolism, muscle contraction, calcium signalling, nucleic acid binding and the spliceosome. The authors' findings identified major highlighted proteins and pathways related to intramuscular fat, free amino acids, and muscle and hair follicle development, which may provide comprehensive insights into meat quality and muscle development across a broad age spectrum.

Kind, and I wish you continued success, as the results of research in the field of proteomics are very important for the development of livestock breeding.

In my opinion, I recommend the submitted article for publication. The original manuscript by title: Protein Profiles for meat characters and Development of thigh meat at different ages of chicken, submitted to the Special Issue: Applications of Proteomics in Food Technology of the Foods journal, meets all the requirements for this type of submission.
  Many factors affect the characteristics of meat. However, the pathways and proteins related to the effect of age on meat quality and muscle development have not been studied so far. Therefore the research results are pioneering. The present study aimed to elucidate the complicated relationship between these features using mass spectrometry with liquid chromatography at three different ages. Proteomics studies included a total of 341 differential expressed proteins (DEP) (fold change ≥ 1.50 or ≤ 0.67 and p < 0.05) among 45 samples of thigh meat (15 pieces per age) of Beijing-You (BYC) chicken, collected aged 150, 300 or 450 days respectively (D150, D300 and D450). The authors analyzed the network of protein interactions and the Markov cluster (MCL) algorithm; 91 DEPs were divided into 26 MCL clusters, which were linked to lipid transporter activity pathways, nutrient reservoir activity, PPAR and MAPK signalling pathways, focal adhesion, ECM-receptor interaction, cycle cell, oocyte meiosis, ribosome, taurine and hypotaurine metabolism, glutathione metabolism, muscle contraction, calcium signalling, nucleic acid binding and spliceosome. The authors' findings have identified major highlighted proteins and pathways related to intramuscular fat, free amino acids, and muscle and hair follicle development, which can provide comprehensive insights into meat quality and muscle development across a broad age spectrum.

Reviewer 5 Report

First of all, I congratulate the authors for this interesting work. I think it would be better for the authors to state the novelty of this study in more detail, unlike previous similar studies.

Round 2

Reviewer 2 Report

The manuscript can be accepted for publication.